# No impact of performance-based financing on the availability of essential medicines in Burkina Faso: A mixed-methods study

Julia Lohmann [1,2‡]*, Stephan Brenner [2‡], Jean-Louis Koulidiati[2], Serge M. A. Somda [3,4], Paul Jacob Robyn[5], Manuela De Allegri[2]

**1** London School of Hygiene & Tropical Medicine, London, United Kingdom, **2** Institute of Global Health, Heidelberg University Hospital and Medical Faculty, Heidelberg, Germany, **3** Centre MURAZ, Bobo-Dioulasso, Burkina Faso, **4** UFR/SEA, Université Nazi Boni, Bobo-Dioulasso, Burkina Faso, **5** Health, Nutrition and Population Global Practice, World Bank, Washington, DC, United States of America

‡ JL and SB share first authorship on this work.
* julia.lohmann@lshtm.ac.uk

**Data Availability Statement:** The quantitative datasets analyzed in the current study is available upon request in the World Bank's Central

## Abstract

Access to safe, effective, and affordable essential medicines (EM) is critical to quality health services and as such has played a key role in innovative health system strengthening approaches such as Performance-based Financing (PBF). Available literature indicates that PBF can improve EM availability, but has not done so consistently in the past. Qualitative explorations of the reasons are yet scarce. We contribute to expanding the literature by estimating the impact of PBF on EM availability and stockout in Burkina Faso and investigating mechanisms of and barriers to change. The study used an explanatory mixed methods design. The quantitative study component followed a quasi-experimental design (difference-in-differences), comparing how EM availability and stockout had changed three years after implementation in 12 PBF and in 12 control districts. Qualitative data was collected from purposely selected policy and implementation stakeholders at all levels of the health system and community, using in-depth interviews and focus group discussions, and explored using deductive coding and thematic analysis. We found no impact of PBF on EM availability and stockouts in the quantitative data. Qualitative narratives converge in that EM supply had increased as a result of PBF, albeit not fully satisfactorily and sustainably so. Reasons include persisting contextual challenges, most importantly a public medicine procurement monopoly; design challenges, specifically a disconnect and disbalance in incentive levels between service provision and service quality indicators; implementation challenges including payment delays, issues around performance verification, and insufficient implementation of activities to strengthen stock management skills; and concurrently implemented policies, most importantly a national user fee exemption for children and pregnant women half way through the impact evaluation period. The case of PBF and EM availability in Burkina Faso illustrates the difficulty of incentivizing and effecting holistic change in EM availability in the presence of strong contextual constraints and powerful concurrent policies.

Microdata Catalogue. Baseline: https://microdata.worldbank.org/index.php/catalog/2761 Endline: https://microdata.worldbank.org/index.php/catalog/3504 The qualitative data analyzed are not publicly available as they cannot be sufficiently anonymized. Many of the interviews are with higher-level stakeholders who are likely identifiable from their responses, and confidentiality therefore cannot be guaranteed.

**Funding:** This study was funded by The World Bank through the Health Results Innovation Trust Fund (HRITF), through grants to the Heidelberg University Hospital (principal investigator: MDA) for overall scientific coordination as well as for the qualitative study, and for quantitative data collection to Centre MURAZ, Burkina Faso (principal investigator: Hervé Hien). The funders had no role in study design, data collection and analysis, decision to publish, or preparation of the manuscript.

**Competing interests:** We have read the journal's policy and the authors of this manuscript have the following competing interests: This work was supported by The World Bank through the Health Results Innovation Trust Fund (HRITF). The World Bank was engaged in the overall design of the intervention and the impact evaluation (IE), but had no role in data collection, data management, data analysis and interpretation, preparation, review and approval of the manuscript. MDA was the Principal Investigator of the IE, but received no direct compensation from the World Bank. JL, SB, and JLK were fully or partially funded by the HRITF grant to the University and worked on the IE (data collection, management, and reporting to the World Bank), but received no direct payment by the Bank nor any compensation for manuscript preparation (which occurred outside the framework of the contract with the World Bank). SMAS is an employee of Centre MURAZ, but not directly funded by the HRITF grant allocated to Centre MURAZ for data collection. PJR is a World Bank employee and Co-PI of the impact evaluation, but participated in writing this paper independently of his professional engagement. The views reported in this paper represent the views of the authors exclusively and not those of the funding agency.

## Introduction

Access to safe, effective, and affordable essential medicines is critical to both the provision of quality health services and to financial risk protection [1]. The Sustainable Development Goals therefore promote access to essential medicines (EM) as central element toward universal health coverage [2]. While the World Health Organization's (WHO) Essential Medicine List has been widely adopted and adjusted to national contexts by many countries, availability of EM varies greatly across specific disease or patient groups (e.g., chronic illnesses, child formulas, reproductive health), especially in low- and middle-income countries (LMIC) [3, 4].

Shortages of EM are often linked to factors related to the global pharmaceutical market (e.g., price fluctuations), manufacturing (e.g., national production capacities), supply chain (e.g., consumption reporting and procurement structures), and/or the political environment (e.g., regulatory frameworks) [5]. Especially in LMIC, however, managerial inefficiencies, communication challenges, and limitations in transparency, autonomy, and accountability often contribute significantly to supply chain interruptions [6, 7].

Innovative financing mechanisms such as performance-based financing (PBF) have been recommended and piloted in several LMIC to approach inefficiencies in health service delivery, including the availability of EM [8]. A recent systematic review concludes that PBF has tended to improve availability of medicines and stockouts of vaccines, both for targeted and untargeted items, but evidence is generally of low certainty [9]. However, such beneficial effects on EM supply are not consistently observed across PBF schemes [e.g., 10, 11]. The little available qualitative evidence points at the importance of specific PBF design and implementation context [12]. For stockouts of medicine and availability of vaccines, the systematic review concludes that evidence is lacking and/or unclear [9]. Further understanding of the role of PBF schemes in improving availability of EM in LMIC is therefore needed.

Combining quantitative and qualitative data collected within the framework of a larger impact evaluation, this study assesses the effect of a PBF intervention in Burkina Faso on the availability and the frequency of EM stockouts among primary level health facilities, contributing further insight into to successes and limitations of PBF in regard to availability of medicines and beyond.

## Contextual background

### Study setting

Burkina Faso is a landlocked low-income country in West Africa. At the onset of our study in 2013, it had a population of about 17 million with about 73% of the population living in rural areas [13]. In 2013, total per capita health expenditure was estimated at 6.3% of GDP [13] The population faces a high degree of morbidity and mortality due to both communicable and non-communicable diseases [14]. In 2013, child and maternal mortality rates were high with 95.8 under-five deaths per 1,000 live births and 362 maternal deaths per 100,000 live births [13]. The public health system is organized in three levels, with primary health care at the community level provided by *Centres de Santé et de Promotion Sociale* (CSPS). Recent health policy reforms have focused on improving access by removing financial barriers for maternal and child health services, most notably partial removal of user fees for delivery and emergency obstetric and newborn care (S*ubvention des accouchements et soins obstétricaux et néonatals d'urgence*, referred to as *SONU* by health providers and policy makers in Burkina) since 2007 [15], which in 2016 was replaced by a full removal of user fees for all services for pregnant and lactating women as well as for all children under the age of five in 2016 (*Gratuité de soins au profit des femmes et des enfants de moins de cinq ans*) [16].

## Medicine supply system

Within the public sector, EM and the majority of other common medicines and consumables are centrally purchased through the *Centrale d'Achats des Médicaments Éssentiels Génériques et de Consommables Médicaux* (CAMEG), a private not-for-profit organization within the Ministry of Health (MoH), and distributed to ten regional warehouses [17]. Within regions, the regional health directorates (*Directions Régionales de la Santé*, DRS) manage the supply the district-level warehouses (*Dépôts Répartiteurs de District*, DRD) across the current 70 districts. At district levels, the district management teams (*Équipes Cadre de District*, ECD) manage the warehouses and supply EM across CSPS and other health facilities.

CSPS are autonomous in managing their CSPS pharmacy, ordering medicines and other consumables from the district according to respective need. They recover costs from service users who purchase medicines and commodities directly from the CSPS pharmacies at an annually determined public price [17]. Medicines provided for free or at reduced cost as part of the *gratuité* are reimbursed or directly provided by the government. Following the implementation of the Bamako Initiative's community financing model, financial management of CSPS revenues (user fees, PBF bonuses, government reimbursement, other subsidies), expenditures (medicine purchases, staff incentives, and other expenses) and savings are handled by local management committees (*Comités de Gestion*, COGES), comprised of community members and CSPS staff.

## Medicine supply situation prior to PBF implementation

The 2012 Service Availability and Readiness Assessment (SARA) indicated that in less than 1% of health facilities, all 14 essential medicines as defined by WHO were available, with antibiotics most frequently available, and medicines for non-communicable disease treatment least available. The SARA further found dramatic regional variation in EM availability [18]. A study based on the same SARA data set, but focusing on the availability of essential medicines for women and children specifically, found that on average across health facilities, between 4 and 5 out of 12 EMs for maternal health services and between 6 and 7 out of 12 EMs for child health services were available [19]. MoH district-level routine data indicated that in 2013, stockouts of tracer medications occurred at one point or another in 25% of district-level pharmacies [20]. While all using different definitions of EM, neither congruent with the list of EM incentivized by PBF and used in our study, pre-implementation evidence strongly suggests substantial gaps in availability of commonly used medication, as well as the implication of all levels of the hierarchy in causing these gaps.

Evidence on patient perceptions and satisfaction with medicine availability in the years before PBF implementation is limited, but the few available studies imply that community members tended to be quite dissatisfied with the frequent non-availability of medicines [21, 22].

## General PBF design and implementation

Between 2014 and 2018, the MoH, with financial and technical assistance from the World Bank, expanded a pilot-PBF scheme from three to an additional twelve districts in six regions purposely selected as having health indicators below the national median at the onset of the intervention [23]. Within each region, the MoH purposely selected two districts for PBF implementation based on particularly poor outcomes on four key indicators: (i) contraceptive prevalence rate; (ii) assisted deliveries; (iii) antenatal consultations; and (iv) postpartum consultations. In each selected district, all CSPS were part of the PBF intervention.

The primary objective of this scaled-up scheme was to improve the utilization and quality of defined packages of health services comprising a broad range of–at CSPS level–basic health

services, including maternal and child health (MCH), reproductive health, general outpatient care, HIV and tuberculosis services [23, 24]. Under the PBF's contractual arrangements, the MoH reimbursed facilities in the form of case-based payments for incentivized services provided. Service unit prices were adjusted to reflect facilities' specific service provision context (i.e. remoteness of catchment population, staffing levels, and distance from the district capital), with higher reimbursement rates for more remote and less accessible facilities. In addition, service quality was purchased quarterly through application of an extensive quality checklist. Facilities exceeding a certain quality threshold were eligible for additional quality bonuses based on their performance, but there were no sanctions for services of substandard quality. At the CSPS level, quantity performance was assessed by an independently contracted verification agency (*Agences de contractualisation et verification*, ACV), while quality performance was verified directly by the respective district management teams. The ACV were also tasked to coach health facilities in how to improve performance.

## PBF design and theory of change in relation to essential medicines

Supply of EM was explicitly incentivized through 19 PBF tracer medicines (see Table 1) expected to be available in CSPS pharmacies at all times and assessed as part of the quality checklist. Availability of PBF tracer EM in the pharmacy contributed less than 1% to the overall quality score on which quality bonuses were based. However, the majority of quality points were to be gained by correct application of treatment protocols for basic health services,

**Table 1. Essential medicines included in the study.**

| Drug preparations | Essential medicines incentivized by PBF | Essential medicines not incentivized |
|---|---|---|
| *Pediatric formulas/dosing* | Amoxicillin syrup | Cotrimoxazole syrup |
| | Oral rehydration salt powder | Metronidazole syrup |
| | EITHER: Artesunate + amodiaquine | Tetracycline eye ointment |
| | OR: Artemether + lumefantrine tablets | Vitamin A capsules |
| | | Paracetamol syrup |
| *Adult formula/dosing* | Amoxicillin tablets | Phenoxymethylpenicillin tablets |
| | Ciprofloxacin tablets | Cloxacillin tablets |
| | Cotrimoxazole tablets | Mebendazole tablets |
| | Erythromycin tablets | Miconazole ointment |
| | Metronidazole tablets | Artesunate + Amodiaquine tablets |
| | Quinine tablets | Artemether + Lumefantrine tablets |
| | Iron + folic acid tablets | Aspirin tablets |
| | Sulfadoxine + pyrimethamine tablets | |
| | Ibuprofen tablets | |
| | Paracetamol tablets | |
| *Injectable preparations* | Ampicillin | Benzathine Benzylpenicillin |
| | Quinine | Magnesium sulphate |
| | Butylscopolamine | Oxytocin |
| | Diazepam | Betamethasone |
| | Furosemide | Hydrocortisone |
| | Glucose (5% OR: 10%) | Normal saline (0.9%) |
| | | Ringer lactate |
| | | Sterile water |
| | | Phenobarbital |
| | | Metoclopramide |

including elements and processes necessitating availability of EM which contributed about 12% to the overall quality score.

Case-based payments were not tied to service quality, so that facilities received quantity reimbursements even if respective medicines were not available and/or administered. However, against the above-described recurrent issues with EM supply and the associated patient dissatisfaction challenges this posed to health facilities and health workers, we assumed that the importance of EM availability in attracting and retaining patients would nonetheless constitute an implicit incentive for health facilities to work on their EM availability.

We further assumed that beyond acting as a motivator, PBF would enable facilities to improve their EM as PBF incentive payments would enhance the overall financial situation of the health facility and thereby allow purchasing of and holding sufficient EM stocks, availability at the CAMEG provided. We also assumed that coaching from the ACV would allow facilities improve both their financial management and drug supply skills, thereby contributing to improved proactive stock management and reduced stockouts.

Finally, important to note is that in regard to medicines, one of the fundamental principles of PBF–procurement autonomy in a free market of accredited suppliers–could not be implemented in Burkina Faso as the government resisted officially authorizing PBF facilities to procure medicines from other accredited wholesalers other than CAMEG.

In joint consideration of the above and of experiences from other countries [9], we expected to find a small positive overall effect of PBF on EM availability. We further expected to find much heterogeneity by organizational units and over time, reflecting differences in understanding, motivation, and contextual circumstances.

## Methods

### Ethics statement

The study received ethical clearance from the Burkina Faso National Ethics Committee (protocol number 2013-7-06) and the Ethics Committee of the Medical Faculty at Heidelberg University (protocol number S-272/2013), and was authorized by the Ministry of Health of Burkina Faso. Prior to data collection, support from regional and district health officers was obtained. Authorization and support from health facility in-charges was obtained prior to quantitative data collection in their health facilities. All qualitative respondents to in-depth interviews provided written informed consent to participate. For community focus group discussions, oral informed consent was sought from the local community leader as well as from focus group discussants in alignment with local research standards.

### Study design

This study employed an explanatory mixed methods design to understand how PBF affected availability of EM in Burkina Faso [25]. Specifically, it used quantitative data to estimate the impact of PBF on availability of EM, and qualitative data to understand why and how–or why not–change came about. To do so, we relied on quantitative data collected as part of a broader impact evaluation of the PBF intervention conducted between 2013 and 2017, and qualitative data collected in 2018 within the framework of a qualitative follow up study to the quantitative impact evaluation. Hereafter, we describe the methodological approaches for the quantitative and qualitative study components separately.

**Quantitative study component.**    *Quantitative design.* The impact evaluation from which we drew quantitative data employed a quasi-experimental pre-test-post-test design with independent controls. We briefly summarize the design and analytical approach below, but would like to refer the reader to the publication by De Allegri et al. (2019) for details [24].

*Quantitative samples.* All 12 intervention districts in the six regions to which PBF was newly scaled up in 2014 were included in the evaluation. As control districts, PBF implementers identified two additional districts without PBF per region, judged as comparable to the intervention districts in terms of health indicators and service provision structures. The study sample consisted of all CSPS in the twelve PBF districts and a random sample of CSPS in the twelve control districts. This resulted in total of 523 CSPS, surveyed at both baseline and endline for a fully balanced panel. Of note, the decision to assign the PBF intervention at district rather than at CSPS level within districts was taken by local decision makers for feasibility reasons given the context.

*Data collection.* Data were collected immediately before (October 2013–March 2014) and about three years after (April–June 2017) implementation start (i.e., baseline and endline). At each time point, a comprehensive facility-based survey was conducted, which also included a health facility inventory checklist. This checklist included indicators measuring the availability and frequency of recent stockouts for a range of EM, including the 19 PBF tracer medicines in Table 2. Data collection teams spent one day at each sampled CSPS facility, collecting data on paper at baseline and with android mobile devices at endline.

*Outcome variables.* We defined four outcome variables for this study. Based on the information collected in the facility inventory, we first grouped all medicines relevant to the CSPS level for which data was collected into two groups: tracer EM (i.e., the 19 incentivized by the PBF) and non-tracer EM (i.e., 22 non-PBF-incentivized medicines) (see Table 2). For each group, we then created binary variables representing the availability (defined as at least one unit in line with the PBF indicator definition) and the stockout frequency (defined as one or more interruptions of stock during the 30 days preceding each survey) of each medicine included in each group. For 60 CSPS, either baseline or endline data was missing for all medicines included in the analysis due to data collection errors. We therefore excluded these facilities entirely, resulting in a total facility sample of 465. Remaining missing data ranged from 0% to 4% per medicine for availability and from 3% to 30% for the stockout variable. Based on observations of how data collectors filled checklists during data collection supervision visits, we are confident that remaining missing data indicate non-availability or stockout, and replaced these data accordingly (i.e., conservative approach to replacing missing data). As a robustness check, however, we also calculated an "optimistic scenario" where we inversely assumed

**Table 2. Qualitative sample.**

| Stakeholder group | Number of respondents | Data collection method |
|---|---|---|
| Implementing agencies within the MoH and auxiliary agencies, including the ACV | 21 | In-depth interviews |
| Regional and district health managers of PBF regions and districts | 12 | In-depth interviews |
| CSPS in-charges purposely selected for maximum variation in facility size and remoteness and observed change during the PBF implementation period | 22 | In-depth interviews |
| COGES presidents from the selected CSPS | 7 | In-depth interviews |
| Focus groups of 8–12 community members from 15 villages in the CSPS catchment areas | 158 | Focus group discussions |
| Community health workers from the selected villages | 7 | In-depth interviews |
| Key informants at national level (MoH technical and financial departments) | 10 | In-depth interviews |

availability and no stockout. Resulting impact estimates differed slightly in magnitude, but not in direction and statistical significance, so that we are confident in the results obtained with the conservative imputation and will only present those in the following. Following data cleaning, we created four composite scores for each outcome variable (i.e., availability of tracer EM, availability of non-tracer EM, stockout of tracer EM, stockout of non-tracer EM) by additive aggregation of the respective binary variables.

*Data analysis.* We employed a Difference-in-Differences (DID) approach to identify the impact of PBF compared to status quo for each of the four outcome variables as expressed by the following regression equation:

$$Y_{dft} = \alpha_f + \beta \cdot 2017_t + \delta \cdot [PBF_d * 2017_t] + \epsilon_{dft},$$

where $Y_{dft}$ is the outcome variable for CSPS f in district d at time t with t = {2014, 2017}. $2017_t$ is a dummy variable indicating endline observations, thus coefficient β gives the time fixed effect. $PBF_d$ is a dummy variable that equals one for CPSP in PBF districts and zero for CPSP in control districts. $\alpha_f$ are CSPS fixed effects, and $\epsilon_{dft}$ is the error term. Standard errors were clustered at the district level as the level of treatment assignment [26]. The coefficient δ gives the DID estimate for the effect of being located in a PBF district when compared to non-PBF districts. Our analysis is therefore challenged by a relatively low number of clusters (24 districts), which might lead to the estimation of downward-biased standard errors with an over-rejection of the null hypothesis [27, 28]. We addressed this threat to validity by using a 'wild bootstrap' to adjust confidence intervals [28, 29]. Unfortunately, routine data on medicine availability to test for parallel trends pre-intervention are not available. However, an interrupted time series study on the impact of PBF in Burkina Faso on various health care utilization indicators showed no difference in pre-intervention trends for the majority of investigated EM-dependent services, supporting our assumption that this was the case for EM availability as well [30].

**Qualitative study component.** *Qualitative design.* Qualitative data were drawn from a cross-sectional study conducted in 2018 as a follow-up to the larger impact evaluation to better understand impact evaluation results, specifically what had allowed PBF to produce a positive impact on some indicators of service quality and utilization, but not on others. Data collection tools included specific questions regarding medicine availability, having emerged as an issue of interest in the quantitative impact evaluation and ensuing dialogue with implementation stakeholders.

*Qualitative samples.* Respondents were purposively sampled to represent key stakeholder groups directly or indirectly involved in PBF, as detailed in Table 2.

Given high staff turnover in light of the fact that qualitative data collection took place one year after the quantitative endline, we took care to select and/or trace respondents who had been in office during the impact evaluation study period.

*Data collection.* Data were collected in September 2018, four and a half years after the start of the intervention and one and a half years after the impact evaluation endline. PBF was still officially on-going at the time of data collection, but activities including incentive payments had been on hold due to financial issues. Data were collected through in-depth interviews (focus group discussions with community members given focus of the overall qualitative study), by the first and second author with support of trained interviewers along semi-structured interview guides. Interview guides for all stakeholder groups contained question related to role of and impact on EM availability. Interviews and focus group discussions were recorded, verbatim transcribed, and where not already conducted in French, translated into French.

*Data analysis.* Transcripts were coded by the first and second author using NVivo, and tri-angulated by MDA using a largely deductive thematic coding approach, with few additional emerging codes added as we proceeded through the material. For the purpose of this paper, the first author revisited all initial codes relevant to EM availability (including decision-making processes at CSPS level; observed changes in quality; change mechanism regarding purchasing medicines and supplies; factors affecting change in supply chains, understanding of PBF, leadership, financial aspects, *gratuité*, other factors) and translated material into a framework matrix [31], summarizing statements and key quotes per code for each respondent or focus group.

### Triangulation of quantitative and qualitative results

Three of the authors independently interpreted the material contained in the qualitative framework matrix in relation to the results obtained from the quantitative analysis, arriving at highly similar conclusions. They then discussed the emerging interpretations with the other co-authors for a final interpretation and conclusion.

## Results

### Quantitative findings

Of the 465 CSPS in the sample, 366 facilities received PBF and 99 served as controls. All CSPS in both study arms were publicly owned. Control sample size was substantially smaller, as in control districts, a random sample was taken for a 3:1 intervention to control ratio.

As shown in Table 3, at baseline the number of available tracer EM and non-tracer EM was slightly higher at PBF CSPS compared to control CSPS (variables 1 and 2). At endline, EM availability decreased slightly more at PBF CSPS compared to control CSPS. We observed a slightly higher degree of recent stockouts (variables 3 and 4) at PBF CSPS compared to control CSPS. At endline, however, the observed number of EM with stockouts was slightly higher at control CSPS compared to PBF CSPS.

Table 4 shows the estimated PBF effect sizes for each outcome variable. We did not observe any impact of PBF on EM availability or stockouts.

### Qualitative findings

Unlike indicated by the results of the quantitative analyses, in the qualitative study, respondents did report improvements in EM supply in the course of the PBF implementation period and explained the mechanisms through which improvements were realized. Specifically,

**Table 3. Descriptive results for each outcome variable by study arm and time point.**

| | | Baseline | | | | Endline | | | |
| | | (2013/14) | | | | (2017) | | | |
| | | Control CSPS | | PBF CSPS | | Control CSPS | | PBF CSPS | |
| | | (n = 99) | | (n = 366) | | (n = 99) | | (n = 366) | |
| *Outcome variables* | | *mean* | *(SD)* | *mean* | *(SD)* | *mean* | *(SD)* | *mean* | *(SD)* |
| 1 | Number of tracer EM (19 drugs) with at least 1 unit available in stock. | 15.7 | (2.6) | 16.5 | (2.1) | 15.2 | (2.2) | 15.4 | (2.0) |
| 2 | Number of non-tracer EM (22 drugs) with at least 1 unit available in stock. | 13.7 | (3.3) | 14.9 | (3.3) | 13.8 | (2.7) | 14.6 | (3.0) |
| 3 | Number of tracer EM (19 drugs) with stockout in past 3 months | 2.3 | (2.8) | 3.4 | (4.6) | 3.4 | (2.0) | 3.2 | (2.2) |
| 4 | Number of non- tracer EM (22 drugs) with stockout in past 3 months | 4.8 | (4.4) | 6.6 | (5.6) | 6.7 | (3.0) | 4.8 | (3.2) |

CSPS = Centres de Santé et de Promotion Sociale, EM = essential medicines; SD = standard deviation.

**Table 4. Estimated PBF effect on drug availability and stockouts.**

| | | Effect size | Standard error | 95% CI (wild bootstrap) |
|---|---|---|---|---|
| 1 | Number of tracer EM (19 drugs) with at least 1 unit available in stock. | -0.47 | 0.56 | -0.70, 1.45 |
| 2 | Number of non-tracer EM (22 drugs) with at least 1 unit available in stock. | -0.34 | 0.77 | -0.42, 1.91 |
| 3 | Number of tracer EM (19 drugs) with stockout in past 3 months | -1.44 | 1.09 | -1.49, 0.77 |
| 4 | Number of non- tracer EM (22 drugs) with stockout in past 3 months | -3.69 | 1.86 | -3.93, 0.18 |

CSPS = Centres de Santé et de Promotion Sociale, EM = essential medicines.

narratives converged in that PBF improved EM availability in particular in the first year and a half of implementation, before the introduction of the gratuité policy in mid-2016 exacerbated certain pre-existing challenges.

> *"The gratuité has now put us in trouble, but otherwise, at the time of PBF [alone], we did not have problems of availability of medicines."* (Health facility in-charge)

At the same time, respondents conceded that even before the introduction of the gratuité, PBF could not fully achieve and sustain the desired improvements.

Hereafter, we first describe mechanisms for the positive impact PBF was perceived to have effected, before we detail barriers to more profound and lasting change. Supportive and illustrative quotes are provided in Table 5.

**Mechanisms of change.** As outlined in the introduction, the assumption was that PBF would motivate facilities to work on EM availability both through the explicit incentives as part of the quality verification and reimbursement and through the implicit incentive of attracting and retaining patients with services of high quality, including availability of medicines.

*Explicit incentives for quality improvement*. Narratives indicate that the incentives for EM stock improvements built into the PBF quality component were not particularly powerful. Given the comparatively small amount of money to be gained by quality improvements, particularly in relation to effort and necessary investments, and in light of no sanctions for poor quality, many health facilities appeared to have contented themselves with the seemingly "effortless" earnings through the PBF quantity component, at least initially.

*Implicit incentives for quality improvements*. Initially, few facilities appeared to have realized the implicit incentive linked to offering services of high quality, namely that services of high quality would attract patients, while services of low quality would deter patients, particularly in an environment of enhanced competition in response to PBF. Over time, however, it appears that more and more facilities developed understanding of the importance of service quality in sustaining and increasing service use.

*PBF impact on facility finances*. In principle, CSPS should have been able to maintain sufficient stocks without PBF as they recovered expenses through user fees or–in the case of user fee exempted services–through reimbursements from the government. Interviews at health facility level, however, revealed that prior to PBF, EM stockouts had often been related to financial difficulties nonetheless. Respondents pointed in particular to the partial user fee removal policy for deliveries (SONU), citing difficulties with government reimbursements.

**Table 5. Supportive respondent quotes.**

| **Mechanisms of change** | |
| --- | --- |
| Explicit incentives for quality improvement | *It was more the quantity than the quality [that facilities focused on] because the quality bonus was low, it represented only 25% of the quantity bonus. And also, the quality bonuses need to be used for improving the conditions of care, equipment, consumables, the actors didn't have too much of a direct interest in this.* (Central level respondent) |
| Implicit incentives for quality improvements | *If you don't have quantity, you're not going to get a lot of subsidies. And if you have a lot of subsidies and you don't have quality, you will lose your patients. So the two must go together, so that your CSPS can grow.* (Health facility in-charge) |
| PBF impact on facility finances | *Before PBF, there was the SONU. The SONU . . . for deliveries, the woman paid 900 FCFA and the rest was covered. For two years, 2014 until 2016, we didn't have any reimbursements [from the government]. There was a time when the CSPS was falling apart because there was not enough money to pay for the medicines. When PBF arrived in the district . . . we will never be able to spoil the name of PBF because it was helping to save the services. The money that we earned, instead of motivating the health workers–there was no problem–, the money from PBF went to medicines.* (Health facility in-charge) |
| | *PBF allowed us to live well though the gratuité in its beginnings, because there was a problem with the transfer of the funds. And thanks to the funds we have from PBF, we were able to work without problems, in contrast to districts where PBF does not exist.* (Health facility in-charge) |
| **Barriers to profound and lasting change** | |
| Monopoly of the CAMEG limiting financial autonomy unchanged with PBF | *There was a battle [during PBF design, they] said: we must let go of CAMEG. But they didn't know that the director can't do that, someone put him there . . . If there is a fight to be waged over CAMEG, it has to go to the level of the minister, and CAMEG is not managed by the Ministry of Health alone, there is the Ministry of Commerce, the Ministry of Finance, there has been trouble with this for a long time. So it's not straightforward!* (Regional director) |
| Cumbersome process of procuring from outside the CAMEG only slightly easier with PBF | *If there are stockouts, what we do is, the District Health Officer informs their Regional Director, who then needs to inform central level, the Director General of Pharmacy, Drugs, and Laboratory (DGPML), who analyses and then gives you feedback and the authorization . . . and who will draft a letter for signature by the Secretary General, who then passes the letter back down through the hierarchy, it takes a crazy amount of time. In the end, we tried to make things smoother, [. . .] all within 72 hours.* (Central level respondent) |
| | *You ask for authorization to procure medicines from an approved supplier other than the CAMEG. When you do this once or twice, it might work, but if you continue to ask, those who grant approval at some point are afraid themselves.* (District Health Officer) |
| Inefficient proactive medicine stock management persisting with PBF due to implementation issues | *We were supposed to have a massive presence in the field, but that was not the case because we were given a limited number of days. So that massive presence was missing . . . Because if there are findings [related to quality issues], someone has to be there all the time for the change. I'm not going to blame all on the health system, it must be said that our support role also has been missing a little.* (ACV member) |

(*Continued*)

**Table 5.** (Continued)

| | |
|---|---|
| Perceived unfairness in the quality verification process | *The system to evaluate the level of quality in order to obtain subsidies. . . the was a problem with certain indicators including the medicine stockout rates. I tried to understand why there was a stockout in a CSPS. The health facility in-charge pulled out his order forms and showed that they had not been supplied. I checked with the medicine depot at the district level. They too pulled out their order forms and had not been supplied. So the shortage was national. The evaluators had put zero [points in the quality verification checklist]. I say that this is not how it should be, because we should put "not applicable". We need to build a model that allows health workers to make the effort and then get the reward. But if health workers experience that no matter how much effort they are going to put in, they are going to get zero, there's no point. (Central level respondent)* |
| Frustrations with PBF payment delays | *Some health facilities did not initiate major changes because there was a delay in funding. From the beginning, there were certain rules that PBF was supposed to respect, notably the [financial] motivation. When people work, they expect to be motivated, and it has to be on a regular basis, but that was not the case here. You could go a year or months before you got your money, so there was no motivation. But if the payments were made on a regular, continuous basis, the original motivation would have remained. You know you're doing this, at the end of the month there's your money. But if you work and you wait six months to get the money for what you did—can you imagine?! (District Health Officer)* |
| **Heterogeneity in PBF impacts on EM availability** | |
| Engagement of managers | *It often comes down to the commitment of the health facility in-charges. [. . .] There are some who have taken to PBF and said, this helps us achieve our goals. And indeed, the people who have understood that, they have put in their efforts, they have put in their energy, they have put in their resources, they have involved their COGES. You can see that these people have nothing to do with those who are inactivist, where you see cobwebs, who do as little as possible. (ACV member)* |
| | *If you are unlucky to have a COGES that is not [dynamic], it is difficult for you to move forward even if you want to move forward. So those who have moved forward are people [. . .] with a good COGES that listens when you discuss a problem and give you the resources you need to solve your problem, with the medicines available and all this. [. . .] There are COGES that say: we cannot take the money every time to come and give it to the health workers who will share it and we have nothing. So when it comes to getting resources for the health center, there are some who are not motivated. (Central level respondent)* |
| Facility size | *What makes the CSPS work is the sale of medicines. You pay one million for medicines and you sell them for one million two hundred thousand. You have a profit of two hundred thousand francs and you can take these for your ordinary expenses, to pay the guards, those who clean the CSPS, to be able to function. This is when you have the money to go and buy medicines. If you run out in a week or two, you go and pay, so there is always a profit. But a small CSPS that does not have enough money, they take all their money to pay for the medicines. And when the gratuité does not reimburse, they do not have any more medicines to sell even though the ordinary expenses are there. The small CSPSs should not have the same reimbursement methods as the large CSPSs. A health center that is large enough can go a year without asking for reimbursement knowing that it will come one day. But the small health centers need the reimbursements to be made regularly every two months. (Health facility in-charge)* |

*(Continued)*

**Table 5.** (Continued)

| Availability of alternative suppliers | *The problem here is that we are essentially a rural area, CSPS have no agencies other than the CAMEG nearby. So for us, it is more efficient to order from the CAMEG rather than elsewhere. But in urban districts, since there are other agencies available, it's manageable [to procure from outside the CAMEG].* (Regional Health Officer) |
| --- | --- |

The cash inflow through PBF helped facilities overcome these difficulties and allowed them to procure medicines in sufficient quantities.

In a similar fashion, the nationwide introduction of the gratuité in mid-2016 put many facilities throughout the country in financial trouble. Patient numbers increased dramatically, leading to higher medicine consumption. At the same time, the gratuité program, supposed to prefinance expected expenses for forgone user fees and medicine charges, incurred payment delays. The resulting cash flow challenges left many health facilities across the country unable to pay their medicine bills at times, often resulting in stockouts. PBF facilities were generally less affected as they tended to be more liquid thanks to PBF and could more easily bridge periods waiting for gratuité payments to arrive.

**Barriers to profound and lasting change.** Despite perceived improvements in EM availability as a result of PBF, quantitative findings indicate that these were not sustained over time. Narratives converge on four main elements, which PBF was not or only partly able to change: Challenges related to the monopoly of the CAMEG in the medicine supply chain, inefficiency in proactive stock management, perceived unfairness of the quality verification process, and frustrations with PBF payment delays.

*Monopoly of the CAMEG limiting financial autonomy unchanged with PBF.* As mentioned above, EM purchasing and supply for the public sector is centrally managed by the CAMEG; facilities and districts are not authorized to procure from outside the CAMEG unless under very specific circumstances and with explicit MoH permission. For health facilities, this poses supply challenges in the event that CAMEG is unable to keep certain medicines in stock. Respondents reported that central-level EM stockouts happened from time to time during the PBF implementation period, but were particularly pronounced following the introduction of the gratuité in mid-2016. At CSPS level, this led to stockouts of significant proportions of EM at various points in time, significantly hindering facility operations.

Respondents deplored that the monopoly of the CAMEG limited the extent to which PBF could unfold its key principles of free market and financial autonomy of health facilities, which would have been pivotal in allowing facilities to improve their EM supply situation.

Regional directors reported the CAMEG monopoly having been subject of discussion during the design and implementation stages of PBF, but that fundamental changes were not possible for political and administrative reasons, including that the CAMEG is managed by multiple ministries and has a strong lobby within the health sector.

*Cumbersome process of procuring from outside the CAMEG only slightly easier with PBF.* Respondents conceded that in instances where the CAMEG faced stock outs, facilities were in principle allowed to procure medicines and supplies outside the CAMEG. The required procurement process to do so, however, was perceived as rather lengthy and complex, as facilities first needed to obtain administrative approvals from all levels of the procurement hierarchy. Some respondents, however, pointed out that in response more frequent requests to purchase from outside the CAMEG from PBF districts, supportive provisions within the hierarchy to speed up this approval process were put in place over time as requests.

Several respondents pointed out, however, that despite this provision, MoH administrators tended to be rather reluctant in approving too many requests for outside procurement given the powerful position of the CAMEG.

*Inefficient proactive medicine stock management persisting with PBF due to implementation issues.* One central level respondent pointed out that the CAMEG should not take all blame, but that they have been very reliable in supplying the health system with the exception of the crisis in 2016, after the introduction of the gratuité. Rather, they described how districts and health facilities themselves are oftentimes part of the stockout problem, for instance by not sufficiently anticipating that supply processes take time in the Burkinabè system, with or without PBF, or because of other shortcomings in medicine stock management.

Deficiencies in stock management in many CSPS were also observed by members of the ACV, who in addition to the quantity verification were tasked with supporting health facilities in identifying areas for quality improvement and effecting change. Specifically, ACV observed both stockouts and expiry of EM in consequence of facilities' not anticipating their EM needs well when placing orders. In the context of PBF, health facilities were able to receive coaching in proactive stock management from the ACV. Respondents for the ACV admitted, however, that they generally barely acted on their coaching function as they used almost all of their time and budget to effect the quantity verification, conceding that this might have contributed to the lack of change in service quality.

*Perceived unfairness in the quality verification process.* In the above-described instances where facilities were unable to procure EM despite having the financial means to do so, frustrations were compounded by perceived unfairness in the verification process, leading to an erosion of motivation to make an effort to improve EM availability in the context of PBF. Specifically, respondents reported that PBF quality verifiers would rate facilities down for EM stockouts, even if these stockouts were not of their own fault, but rather due to higher-level supply chain challenges.

*Frustrations with PBF payment delays.* Further, the PBF scheme incurred quite severe payment delays at various points throughout the implementation period. Health workers were frustrated with the non-appearance of the anticipated recognition of effort as well as of anticipated cash to realize planned quality improvements. For many, this compromised motivation to make particular efforts particularly towards quality improvement, particularly where financial investment was necessary such as for EM stock improvement.

**Heterogeneity in PBF impacts on EM availability.** Finally, respondents conceded that beyond the above-cited general mechanisms and issues, there was significant heterogeneity in how facilities and districts approached availability of EM in the context of PBF. Respondents highlighted three main factors: Engagement of managers at all levels, facility size, and availability of alternative suppliers.

*Engagement of managers.* First and foremost, respondents underlined that whether districts and facilities were able to make progress in EM availability and other qualities depended largely on the engagement, energy, leadership capacity, and motivation of the health facility in-charge to effect change with the enhanced means PBF offered to them.

Respondents further noted the important role of the District Health Officer and their engagement in enabling EM availability, and pushing quality improvement in the district more generally. Findings for instance revealed that District Health Officers handled issues with stockouts at the CAMEG in very different ways, with some being very reluctant to approve and pass on requests from health facilities due to the administrative burden and political dimension, and others being more supportive in either requesting official approval or allowing facilities the purchase of missing drugs from private pharmacies even in the absence of official approval.

Finally, respondents underlined the importance of cooperation and engagement of the COGES in improving EM availability, as it is the COGES that manages the health facility finances. It appears that understanding of PBF and reactions to PBF varied tremendously from COGES to COGES, with some realizing its potential in improving their CSPS for the better, while others were envious of the "gift" to health workers they had no part in–as COGES members were not entitled to individual financial incentives in the context of PBF. Ability of health facility staff to engage their COGES in the context of PBF was therefore reported as one important success factor.

*Facility size*. In relation to the above-discussed financial capacity issues, health facility in-charges pointed out that the size and frequentation of the facility had a major impact on whether facilities were able to cope with the payment delays in financial terms, with direct impact on EM availability. Respondents explained that the sale of medicines constitutes the primary mechanism of generating revenue for health facilities, as medicines are sold at a small profit. By principles of scale, large health facilities, turning over large amounts of medicines, are able to generate much larger revenue in absolute terms, allowing them to build up savings whereas small facilities spend their entire revenue on ordinary expenses.

*Availability of alternative suppliers*. Finally, in relation to the possibility of buying outside the CAMEG, respondents pointed out that even in the presence of approval, private pharmacies or other medical retailers are usually only available in urban centers, while more remotely located districts and CSPS often fully depend on the centralized system in absence of alternative procurement routes.

## Discussion

Our study contributes to expanding the yet small body of evidence on the impact of Performance-based Financing on the availability of essential medicine, and more generally highlights the interplay between explicit incentives, implicit incentives, and contextual barriers in the implementation of complex interventions. Using both quantitative and qualitative data, we estimated the impact of PBF on EM availability and stockouts and explained mechanisms of change–or lack thereof. We summarize findings in relation to the theory of change presented earlier in Fig 1.

Unlike a recent review [9], which identified largely positive effects on EM availability in existing PBF impact evaluations, our quantitative analysis revealed no impact of PBF on EM availability in Burkina Faso approximately three years into implementation. Qualitative accounts, however, indicated that EM availability had initially increased as a result of PBF, due to explicit and implicit incentives for improvement as well as improved facility finances. However, even during the initial period, qualitative reports indicated that PBF impact fell short of enhancing EM availability to fully sufficient levels. The introduction of the gratuité policy in mid-2016 dramatically increased demand for medicines and came with its own implementation challenges, thereby introducing further barriers to change. Our findings in regard to why PBF did not lead to sustained change in EM availability resonate with findings from other contexts, illustrating how various issues kept health facilities from being able ("can do") and motivated ("will do") to effect more profound change. First, our findings underline the key importance of the implementation context. Against powerful national strategies with their own implementation challenges and rigid medicine supply chain structures, most notably the government's decision to retain the CAMEG monopoly which largely prevented PBF from acting through its key principle of purchasing autonomy, facilities were limited in their effective ability to induce and sustain change in the context of PBF. This is not surprising in light of many prior PBF experiences having found lack of financial and management autonomy being

| Theory of change | Summary of findings |
|---|---|
| *EM-related elements of the quality checklist motivate health facilities to improve EM availability* | Quality incentives not very powerful against „effortless" earnings through (quality-unconditional) service quantity<br><br>Facilities lack motivation to make an effort given perceived unfairness of the quality verification process in relation to EM and payment delays |
| *Role of EM availability in patient satisfaction motivates facilities to improve to attract and retain patients* | Potential of EM availability in enhancing facility earnings through quantity incentives recognized by few initially, but more so over time |
| *PBF enhances facility's financial situation, allowing them to purchase sufficient EM stocks* | PBF acted as a buffer to negative impact of the SONU and gratuité on facility finances, allowing continued EM purchases<br><br>PBF payment delays limited the impact of PBF on facility finances |
| *ACV coaching enhances facilities' financial and proactive stock management capacity* | Coaching barely implemented due to ACV capacity constraints |
| *Facilities anticipate, order and stock sufficient EM quantities* | Supply issues at the CAMEG hindered facilities from ordering sufficient quantities of EM particularly after the introduction of the gratuité<br><br>CAMEG monopoly remained unchanged for political reasons; purchasing outside the CAMEG was possible but extremely cumbersome |
| *Increased availability and decreased stockouts of EM* | No overall PBF impact on EM availability and stockouts<br><br>Heterogeneity by engagement of managers, facility size, and availability of alternate suppliers |

**Fig 1. Study findings in relation to theory of change.**

obstructive to more profound impact on EM availability, as highlighted by a recent realist review of the literature [32]. In contrast, the example of Cameroon shows how an explicit liberalization of the medicine supply system in the context of PBF can substantially improve the supply situation, although not necessarily to the desired extent given the presence of other contextual factors and implementation challenges [12]. Acknowledging this a priori limitation of the Burkina Faso PBF program in regards to medicines, several of our study respondents implied that more profound progress could have nonetheless been achieved with proactive and far-sighted medicine supply management at all levels of the supply chain. In fact, the PBF program anticipated a need for strengthening of management skills particularly at the CSPS level, but related coaching activities could only be realized to a very limited extent.

Second, the above challenges not only limited facilities' ability to effect change ("can do"), but also the extent to which PBF enhanced facilities' willingness to make an effort towards

improved drug availability ("will do"). This was exacerbated by two issues, namely the frequent payment delays and perceived unfairness with the quality verification in case of EM stockouts at the CAMEG. Again, this is not surprising as the demotivating effect of payment delays and perceived unfairness of various aspects are frequent themes in the PBF literature, in Burkina Faso [33] and beyond [32].

Finally and against this backdrop, it appears that the incentive structure aimed at improvement of EM stocks was not particularly powerful. No sanctions were foreseen for substandard quality of care (including EM availability), and facilities were able to obtain full PBF reimbursements for service quantity even if medicines were not available. The incentive envelope to be gained from quality-independent service provision was much higher than that to be gained from quality-dependent service provision, in absolute terms and particularly relative to perceived effort. Considering this, improving quality including EM supply does not appear to have been a priority for many health facilities, and whether a facility undertook efforts towards improved EM supply much depended on the personality and motivation of its leadership as well as a few structural characteristics.

The case of PBF and EM availability in Burkina Faso illustrates the complexity of translating the fundamental idea of PBF–that with the right set of incentives and located in an enabling environment, health facilities under PBF will become entrepreneurs fixing their own problems [34]–, into practice. Beyond the CAMEG monopoly and issues with fidelity of implementation, stronger enforcement of management capacity building elements might have improved facilities' abilities to proactively and creatively manage medicine stocks to prevent stockouts even in light of contextual challenges difficult to address by PBF.

Most importantly, the case highlights the importance of designing complex incentives for comprehensive change in both quality of and access to care, as also reflected in the increasing use of the concept of effective coverage to understand population access to high-quality health care [35, 36]. Appraising the Burkinabè experience together with experiences from other countries seems to suggest that linking quantity and quality incentives by only reimbursing quantity for services of adequate quality might be preferable in regard to improving EM availability. In countries where substandard quality was not penalized by reduced quantity bonuses or other sanctions, impact evaluations tended to find no effect on EM availability (e.g. Burkina Faso, Cameroon [11], Burundi [37, 38], Zambia [39]). In contrast, in countries where quantity bonuses were discounted for poor quality or quality was otherwise prioritized or linked to quantity, impact on EM availability tended to be more positive (e.g. Tanzania [40], Malawi [41], Congo [42]), albeit not universally so (e.g. Benin, [43]).

While straightforward in theory, however, such interlinked incentive systems need to ensure that they do not inadvertently turn into demotivators by punishing facilities for shortfalls in quality not attributable to their (in)actions, as for instance noted in our study as well as in Malawi [44], Benin [45], or Zambia [46]. Even more importantly, this touches upon a different, much discussed, and highly complex set of questions and challenges around defining and measuring quality of care in a comprehensive and balanced way [47], yet within the reasonable levels of effort necessary in routine implementation so as not to drain too much capacity from local actors' core health service delivery roles and/or to preclude any cost-effectiveness [32, 48]. A review of how quality of care is incorporated into various PBF programs highlights vast heterogeneity in the details of PBF design, but suggests that lessons to date are inconclusive and more learning on context-appropriate incentive design and implementation is necessary [49].

The difficulties in incentivizing quality of care are also reflected in discussions around the future of using pay for performance schemes in Burkina Faso following the close of the PBF pilot scheme investigated in this study in 2018. With the government firmly committed to the gratuité and in light of double funding of service quantity at facility level in concurrent

implementation of PBF and the gratuité, the MoH had planned to adjust case-based payments for service quality based on an annual quality evaluation and certification of each health facility [50]. However, challenges and lack of agreement on how to measure quality, both conceptually and operationally, have contributed to lack of implementation to date [51]. These ongoing roadblocks in dialogue related to integration of programs and how to measure and incentivize quality of care will likely result in further missed opportunities for strengthening critical inputs to health service delivery, including the availability of essential medicine.

## Methodological considerations

Our study is limited by a number of methodological challenges. First, randomization at facility level was deemed impossible within the contextual realities, resulting in a study design with only 24 clusters and therefore low statistical power to detect impact. Looking at effect sizes especially for the two stockout indicators and in light of the qualitative findings, it might well be that a better-powered design would have allowed us to ascertain them as significantly different from zero. Second, the outcome indicator measurement, while in line with PBF indicator definition, is somewhat limited in its indication of actual service readiness in that the availability of only one unit of the respective medicine is required for a medicine to be classified as "available"–while realistically, many medicines are used multiple times per day in normal practice. Further, we cannot exclude that some health facilities purposively kept one unit of each EM in stock in case of quality verification, although we are not aware of any such accounts from implementation practice. Third, the qualitative study component was designed to explain findings of the overall impact evaluation and not geared towards EM availability specifically. While we had explicit questions on EM availability in all interview guides, time to discuss issues of EM supply in depth was limited. We therefore cannot exclude to have missed certain aspects. Finally, we wish to draw attention to a more fundamental issue encountered in our research, but also by others [e.g., 11]. As highlighted by the seeming discrepancy between quantitative and qualitative findings, standard quantitative impact evaluations with two measurement time points are limited in their explanatory power in a complex, multi-intervention setting, prone to miss dynamics over time from which much can be learned, and vulnerable to temporary events such as for instance time-limited national stockouts of certain EM. We therefore end with a plea for the routine use of mixed methods design in impact evaluation to not only understand what work, but also how, why, for whom, and in which context.

## Conclusion

Our study contributes to expanding the yet small body of evidence on the impact of Performance-based Financing on the availability of essential medicines by illustrating mechanisms of and barriers to change in a setting where no sustained impact could be detected three years into implementation. The findings are important not only for the learning agenda for PBF programs that continue today, but also for health system reforms more broadly. They highlight the importance of enhancing health facility autonomy, market liberalization for quality essential medicines, strengthening subnational health authorities and supporting decentralized governance of health care delivery, and better integration and alignment of purchasing schemes as part of system strengthening efforts. If reform processes are not able to move the needle in these critical areas, they will more than likely impede achievements towards Universal Health Coverage. The case of Burkina Faso illustrates the complex interplay of enabling and motivating factors necessary for significant and sustained change in the context of PBF and beyond, most importantly the vital importance of true procurement autonomy and integrated incentives for high-quality service provision.

## Supporting information

**S1 Text. Inclusivity in global research.**
(DOCX)

## Acknowledgments

The authors would like to thank Paul-André Somé for organizing and supervising data collection for the qualitative component.

## Author Contributions

**Conceptualization:** Julia Lohmann, Stephan Brenner, Paul Jacob Robyn, Manuela De Allegri.

**Data curation:** Julia Lohmann, Jean-Louis Koulidiati, Serge M. A. Somda, Paul Jacob Robyn, Manuela De Allegri.

**Formal analysis:** Julia Lohmann, Stephan Brenner, Jean-Louis Koulidiati.

**Funding acquisition:** Julia Lohmann, Paul Jacob Robyn, Manuela De Allegri.

**Investigation:** Julia Lohmann, Stephan Brenner, Jean-Louis Koulidiati, Serge M. A. Somda, Paul Jacob Robyn, Manuela De Allegri.

**Methodology:** Julia Lohmann, Stephan Brenner, Manuela De Allegri.

**Project administration:** Julia Lohmann.

**Writing – original draft:** Julia Lohmann, Stephan Brenner.

**Writing – review & editing:** Julia Lohmann, Stephan Brenner, Jean-Louis Koulidiati, Serge M. A. Somda, Paul Jacob Robyn, Manuela De Allegri.

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
