## [Decision Letter · Decision Letter 0]

25 Nov 2021

PGPH-D-21-00763

No impact of Performance-based Financing on the availability of essential medicines in Burkina Faso: a mixed-methods study underlining the importance of context and intervention design

Dear Dr. Lohmann,

Thank you for submitting your manuscript to PLOS Global Public Health. After careful consideration, we feel that it has merit but does not fully meet PLOS Global Public Health’s publication criteria as it currently stands. Therefore, we invite you to submit a revised version of the manuscript that addresses the points raised during the review process.

In revising your manuscript, please pay particular attention to reviewer comments regarding the conceptual justification of the research question and proposed mechanism for impact so that this is clearly laid out in the introduction and referenced in the subsequent discussion of value added by this study and implications for policy.

We look forward to receiving your revised manuscript.

Kind regards,

Hannah Hogan Leslie, PhD

Academic Editor

Journal Requirements:

1. Please include a complete copy of PLOS’ questionnaire on inclusivity in global research in your revised manuscript. Our policy for research in this area aims to improve transparency in the reporting of research performed outside of researchers’ own country or community. The policy applies to researchers who have travelled to a different country to conduct research, research with Indigenous populations or their lands, and research on cultural artefacts. The questionnaire can also be requested at the journal’s discretion for any other submissions, even if these conditions are not met.  Please find more information on the policy and a link to download a blank copy of the questionnaire here: https://journals.plos.org/plosone/s/best-practices-in-research-reporting. Please upload a completed version of your questionnaire as Supporting Information when you resubmit your manuscript.

2. Please update the completed 'Competing Interests' statement, including any COIs declared by your co-authors. If you have no competing interests to declare, please state "The authors have declared that no competing interests exist". Otherwise please declare all competing interests beginning with the statement "I have read the journal's policy and the authors of this manuscript have the following competing interests:"

3. State the initials, alongside each funding source, of each author to receive each grant.

Reviewers' comments:

Reviewer's Responses to Questions

**Comments to the Author**

1. Does this manuscript meet PLOS Global Public Health’s publication criteria? Is the manuscript technically sound, and do the data support the conclusions? The manuscript must describe methodologically and ethically rigorous research with conclusions that are appropriately drawn based on the data presented.

Reviewer #1: Yes

Reviewer #2: Yes

2. Has the statistical analysis been performed appropriately and rigorously?

Reviewer #1: Yes

Reviewer #2: Yes

3. Have the authors made all data underlying the findings in their manuscript fully available (please refer to the Data Availability Statement at the start of the manuscript PDF file)?

Reviewer #1: Yes

Reviewer #2: Yes

4. Is the manuscript presented in an intelligible fashion and written in standard English?

Reviewer #1: Yes

Reviewer #2: Yes

5. Review Comments to the Author

Reviewer #1: The paper is well-written and implements a very pertinent mixed method analysis; something which should be undertaken by more papers focusing on the impact of health system-level interventions such as strategic purchasing and PBF. In particular, the qualitative assessment is very rigorously done and is very insightful.

I would like to make three comments which could strengthen the nuances and the discussion:

-Research question and relevance: A priori, I don't think there is any reason to think that a convoluted purchasing mechanism such as PBF could or should have any impact on the availability of essential medicines at the health facility level, especially in the absence of any specific interventions targeting essential medicine availability (and especially when facility autonomy is not well implemented). Essential medicines only constitute 2 percent of the quality checklist and even then it's a dummy yes/no variable. While the paper is well-done, I'm not exactly sure why a paper specifically on essential medicine availability - which seems to have been only tangentially incorporated as a policy objective of PBF - is relevant. It is clear that PBF is not very effective for improving the availability of essential inputs which should already be a given in any health system - there is no ex ante reason to reward health facilities to possess basic inputs which they should already have. It is unclear as to what percentage of stockouts were due to poor ordering or lack of medicines at the national warehouses to begin with. All of this to say that these caveats should come across more clearly.

-Timing of quantitative analyses and linking changes in essential medicines availability to other indicators incentivized by PBF: It would be interesting to show a time dimension on the change of the indicator on essential medicines, as well as whether the change in this indicator was higher or lower compared to the change in other indicators, both in terms of the quality index as well as any quantity indicators. This would help contextualize the essential medicines indicator further in the context of the PBF program. It would be relevant to show what happened to other indicators in the same period: the paper alludes to quantity indicators improving faster given the incentives, many of which could only improve if essential medicines and commodities at the facilities are available; what is the reason behind this disconnect? Maybe it's just a question of when the indicators on essential medicines were collected, which is less continuous than quantity indicators? This could help clarify the disconnect between the quantitative and qualitative results.

-Focus more on context, change pathways and theory of change: The paper could benefit from a stronger conceptual framework, which has been put forth in a similar evaluation of the Cameroon program (Sieleunou et al 2019) as well as another paper from Cote d'Ivoire (Duran et al 2020) which also used a qualitative approach. It would be good to unpack the specific pathways through which the program in Burkina Faso evaluated in this program could have changed essential medicine availability. This would also help crispen the recommendations on how the program or other health system interventions could be better designed; focusing on facility autonomy, role of regional authorities/strengthened decentralized governance, as well as the integration of diverse purchasing programs (e.g. getting rid of the gratuite program). This should also proactively indicate what is in the decision space of facilities versus not. The discussion or conclusion sections should more proactively indicate implications of these results, both for the future of the PBF program (is this still ongoing? Integrated to a national strategic purchasing program? Why or why not?)

Reviewer #2: Thank you for the opportunity to review this manuscript. I would like to commend the authors for a job well done overall, preparing a well-written and comprehensive investigation into a very important topic (EM availability) in the context of performance-based financing initiatives in Burkina Faso. This paper will certainly make an important contribution to the field by providing a very in-depth analysis of barriers specific to the Burkina Faso context, while also providing important learnings for other countries.

This paper has several real strengths. First, the mixed-methods design is an excellent choice as it allows the researchers to measure changes in EM stock-outs following PBF implementation, as well as explore themes around implementation qualitatively with stakeholders. As the authors note, policy changes mid-way through the evaluation period may have greatly impacted availability of EM in PBF clinics, but this phenomenon was not captured by the quantitative data collection. I believe the authors do a commendable job in addressing this limitation and that it does not hinder the analysis; rather it provides an important learning regarding study design. Secondly, the authors very clearly lay out the quantitative and qualitative approaches used in this study, which is often a real challenge for mixed-methods papers. Lastly, I very much appreciate their approach to handling missing data in the quantitative analysis and checking for robustness of their findings.

In addition to strengths, I do believe the authors could further strengthen their manuscript by revising some aspects of the manuscript, particularly in the qualitative results section. I recommend revising and resubmitting this manuscript for review. I will discuss my recommendations below, first describing major revisions and then describing minor revisions.

Major revisions

1. The introduction would benefit from a clear explanation of the mechanism by which the authors expect PBF to decease stock-outs or increase EM supply (perhaps around lines 66-68). This hypothesized mechanism appears to be referenced in the results section later in the paper (lines 318-321). If this is indeed the researchers’ hypothesis, it should not be included in the results section but in the introduction as suggested. It could then be referred to in the results as to whether the qualitative research disproved or proved their hypothesis.

2. The qualitative data collection section in the methods could be improved by adding some additional detail, specifically, which participants IDIs were conducted with as opposed to FGDs, and why the different methods were chosen per group. The description of the qualitative sample is very long and hard to follow – it might be more impactful to present in table format. Were any other data collected on the qualitative participants other than role (i.e., age, gender, location, etc.?)

While there is a lot of valuable information in the qualitative results section, in its current format it is so long and full of detail that the reader becomes lost. I have two primary suggestions to help with this:

3a. The quotes used, while valuable in that they highlight the voices of the respondents, are overly long. I recommend reviewing the quotes and trimming text to make them more succinct while staying true to the respondents' views; others could be summarized in a sentence or two rather than quoted. A nice rule of thumb may be 1 direct quote per theme. Additional quotes could also be presented in a table.

3b. Again, due to the length of the results section it becomes difficult for the reader to keep all of the qualitative findings in their head at once. A figure summarizing the qualitative themes and how they interact might be a nice component to add.

Minor revisions

1. I suggest changing the title by excluding “underlining the importance of context and intervention design.” It is an interesting discussion point but does not come across as the main takeaway of the article (rather what comes across is that PBF did not increase EM availability).

2. Is a medicine really considered available if there is only 1 unit present? What if 2 patients are present that day and need the same medicine? Are there standard definitions of medication availability? It would be helpful to either have a citation for the definition of medication availability or brief explanation from the authors regarding this choice.

3. Perhaps personal preference, but I suggest using “purposive” or “purposively” sampled rather than “purposely.”

4. I suggest removing the respondent reference IDs from quotes – they do not add anything for the reader.

6. PLOS authors have the option to publish the peer review history of their article (what does this mean?). If published, this will include your full peer review and any attached files.

**Do you want your identity to be public for this peer review?** For information about this choice, including consent withdrawal, please see our Privacy Policy.

Reviewer #1: No

Reviewer #2: **Yes: **Rebecca Lynn West

---

## [Decision Letter · Decision Letter 1]

2 Mar 2022

No impact of Performance-based Financing on the availability of essential medicines in Burkina Faso: a mixed-methods study

PGPH-D-21-00763R1

Dear Dr. Lohmann,

We are pleased to inform you that your manuscript 'No impact of Performance-based Financing on the availability of essential medicines in Burkina Faso: a mixed-methods study' has been provisionally accepted for publication in PLOS Global Public Health.

Best regards,

Hannah Hogan Leslie, PhD

Academic Editor

Reviewer Comments (if any, and for reference):

Reviewer's Responses to Questions

**Comments to the Author**

1. If the authors have adequately addressed your comments raised in a previous round of review and you feel that this manuscript is now acceptable for publication, you may indicate that here to bypass the “Comments to the Author” section, enter your conflict of interest statement in the “Confidential to Editor” section, and submit your "Accept" recommendation.

Reviewer #1: All comments have been addressed

Reviewer #2: All comments have been addressed

2. Does this manuscript meet PLOS Global Public Health’s publication criteria? Is the manuscript technically sound, and do the data support the conclusions? The manuscript must describe methodologically and ethically rigorous research with conclusions that are appropriately drawn based on the data presented.

Reviewer #1: Yes

Reviewer #2: Yes

3. Has the statistical analysis been performed appropriately and rigorously?

Reviewer #1: (No Response)

Reviewer #2: Yes

4. Have the authors made all data underlying the findings in their manuscript fully available (please refer to the Data Availability Statement at the start of the manuscript PDF file)?

Reviewer #1: (No Response)

Reviewer #2: No

5. Is the manuscript presented in an intelligible fashion and written in standard English?

Reviewer #1: (No Response)

Reviewer #2: Yes

6. Review Comments to the Author

Reviewer #1: Thank you for addressing all my comments.

Reviewer #2: (No Response)

7. PLOS authors have the option to publish the peer review history of their article (what does this mean?). If published, this will include your full peer review and any attached files.

**Do you want your identity to be public for this peer review?** For information about this choice, including consent withdrawal, please see our Privacy Policy.

Reviewer #1: **Yes: **Denizhan Duran

Reviewer #2: No
